# Natural Wollastonite-Derived Two-Dimensional Nanosheet Ni$_3$Si$_2$O$_5$(OH)$_4$ as a Novel Carrier of CdS for Efficient Photocatalytic H$_2$ Generation

**Jiarong Ma** [1], **Run Zhou** [1], **Yu Tu** [1], **Ruixin Ma** [2], **Daimei Chen** [1,*] and **Hao Ding** [1,*]

[1] Beijing Key Laboratory of Materials Utilization of Nonmetallic Minerals and Solid Wastes, National Laboratory of Mineral Materials, School of Materials Science and Technology, China University of Geosciences, Xueyuan Road, Haidian District, Beijing 100083, China; mmmjr666@163.com (J.M.); 3003180004@email.cugb.edu.cn (R.Z.); tuyu0803@126.com (Y.T.)

[2] School of Chemical and Environmental Engineering, North China Institute of Science and Technology, Langfang 065201, China; maruixin@126.com

[*] Correspondence: chendaimei@cugb.edu.cn (D.C.); 2003011690@cugb.edu.cn (H.D.)

**Abstract:** Ni$_3$Si$_2$O$_5$(OH)$_4$ rods (NS) were synthesized via a hydrothermal method, employing natural wollastonite as a template. The hierarchical Ni$_3$Si$_2$O$_5$(OH)$_4$ rods exhibited vertically oriented nanosheets, resulting in a substantial increase in the specific surface area (from 2.24 m$^2$/g to 178.4 m$^2$/g). Subsequently, a CdS/Ni$_3$Si$_2$O$_5$(OH)$_4$ composite photocatalyst (CdS/NS) was prepared using a chemical deposition method. CdS was uniformly loaded onto the surface of the Ni$_3$Si$_2$O$_5$(OH)$_4$ nanosheets, successfully forming a heterojunction with Ni$_3$Si$_2$O$_5$(OH)$_4$. The CdS/NS photocatalyst in the presence of lactic acid as a sacrificial agent demonstrated an impressive H$_2$ production rate of 4.05 mmol h$^{-1}$ g$^{-1}$, around 40 times higher than pure CdS. The photocorrosion of CdS was effectively solved after loading. After four cycles, the performance of CdS/NS remained stable, showing the potential for sustainable applications. After photoexcitation, electrons moved from Ni$_3$Si$_2$O$_5$(OH)$_4$ to the valence band of CdS, where they interacted with the holes via an enhanced interface contact. Simultaneously, electrons in CdS transitioned to its conduction band, facilitating hydrogenation. The enhanced performance was attributed to the improved CdS dispersion by Ni$_3$Si$_2$O$_5$(OH)$_4$ loading and efficient photogenerated carrier separation through the heterojunction formation. This work provides new perspectives for broadening the applications of mineral materials and developing heterojunction photocatalysts with good dispersibility and recyclability.

**Keywords:** Ni$_3$Si$_2$O$_5$(OH)$_4$; wollastonite; CdS; photocatalytic H$_2$ production





## 1. Introduction

The continuous reliance on fossil fuels has led to a critical global crisis, prompting an urgent need to explore clean and renewable energy sources as alternatives to conventional fuels [1,2]. Hydrogen (H$_2$) has emerged as a promising substitute, offering inexhaustible, clean, and renewable properties [3]. Photocatalytic water splitting to generate H$_2$ has gained significant attention as a potential solution to address the current energy shortages [4]. Among the various photocatalysts for H$_2$ generation, cadmium sulfide (CdS) has shown promising properties, including an appropriate bandgap width and negative conductive potential [5]. However, one of the major challenges lies in effectively suppressing the recombination of photogenerated carriers to optimize the photocatalytic performance of CdS for H$_2$ generation [6]. To address this challenge, the formation of heterojunctions via coupling CdS with other semiconductors has been explored as an effective approach to enhance electron and hole separation [7–10]. Nevertheless, CdS-based heterojunction photocatalysts often encounter issues related to the difficult recovery and severe aggregation of CdS, limiting their practical application in commercial photocatalytic water splitting for

$H_2$ production. Therefore, a critical focus in the field is the development of a CdS-based heterojunction photocatalyst with excellent recycling performance and a good dispersion of CdS.

The loading of CdS or CdS-based heterojunction photocatalysts onto a specific carrier and their utilization as composite photocatalysts have proven to be effective strategies to enhance the dispersion and recovery performance of photocatalysts, resulting in an improved $H_2$ production efficiency [11,12]. Minerals, in particular, have been garnering attention for these applications as photocatalyst carriers due to their inherent advantages, such as chemical stability, corrosion resistance, and low cost [13]. By leveraging these properties, mineral-based composite photocatalysts offer promising prospects for sustainable and cost-effective $H_2$ production through photocatalytic water splitting [14]. For instance, Najme et al. prepared a composite photocatalyst (CdS-CNP) by loading the synthesized hexagonal CdS onto the surface of clinoptilolite (CNP) [15]. CdS-CNP exhibited a significant photocatalytic performance due to the enhanced dispersion of CdS by the carrier CNP. Li et al. uniformly assembled spherical CdS nanoparticles on kaolinite by a microwave irradiation process in an aqueous solution to synthesize CdS/kaolinite nanocomposites [16]. Nascimento et al. reported the use of diatomite as a carrier to deposit uniformly dispersed CdS nanoparticles through an aqueous reaction [17]. Zhang et al. [18] uniformly deposited CdS nanoparticles onto the surface of lanthanum-doped halloysite nanotubes (La-HNTs). The photocatalytic hydrogen evolution rate of the CdS/La-HNTs reached 47.5 μmo/h. Peng et al. [19] constructed a hydrophilic conductive catalyst support by encapsulating graphene-coated montmorillonite (rGO/Mt). They synthesized rGO/Mt@CdS/MoS$_2$ composite materials via a hydrothermal method, exhibiting a high hydrogen evolution rate. Wang et al. [20] achieved an impressive hydrogen production performance by synthesizing CdS nanorod arrays decorated with Ni-silicate nanosheets (CdS NRS@Ni-silicate) through a simple surface decoration method. However, the majority of the reported mineral carriers are unable to form heterojunctions with CdS. Loading CdS-based heterojunction photocatalysts often leads to complex processes and increased costs, which hinders their commercial application in photocatalytic $H_2$ production. Additionally, the use of block particles as carriers often provides a limited improvement in the dispersion of CdS compared to carriers with larger specific surface areas, such as sheets or rod-shaped carriers. Lin et al. developed all-in-one hybrid photocatalyst Cd/CdS/halloysite nanotubes (P-Cd/CdS/HNTs) by a three-step method for excellent $H_2$ evolution, and it was also pointed out that the HNTs were outstanding carriers for well-dispersed CdS nanospheres, having a greater number of exposed catalytic sites [21]. Thus, the development of a mineral carrier with specific morphological features capable of forming an effective heterojunction with CdS is indeed an innovative and significant approach. Such a carrier would offer the benefits of both enhanced dispersion and efficient photogenerated carrier separation, potentially overcoming the limitations observed in traditional mineral carriers. By combining the advantages of mineral carriers with the formation of heterojunctions, this novel approach holds promise for advancing the commercial application of photocatalytic $H_2$ production, contributing to a cleaner and more sustainable energy future.

The development of inorganic nanoparticle synthesis has enabled the preparation of various morphologies of silicates, such as tubular, sheet, and spherical structures [22–24]. Nickel silicate hydroxide ($Ni_3Si_2O_5(OH)_4$) is a common silicate mineral with a layered structure. The nickel oxide octahedron ($NiO_6$) layer and the silicon oxide tetrahedron ($SiO_4$) layer form a special lamellar structure [25,26]. $Ni_3Si_2O_5(OH)_4$ nanotubes can be synthesized by utilizing the self-curling effect of the layered structure. Yang et al. synthesized $Ni_3Si_2O_5(OH)_4$ nanotubes, which had a layered structure, by the alkalinity-tuned hydrothermal approach at a temperature of around 200 °C [27]. Furthermore, hollow micro/nanostructured $Ni_3Si_2O_5(OH)_4$ can also be fabricated using a template method. Wang et al. prepared $Ni_3Si_2O_5(OH)_4$ nanotubes by a self-templating route using $SiO_2$ nanorods as sacrificial templates and a hydrothermal method [28]. Liu et al. prepared hierarchical hollow $Ni_3Si_2O_5(OH)_4$ microflowers by the one-pot hydrothermal method

using $Ni(NO_3)_26H_2O$ and 1.20 g urea as the raw materials [29]. The hierarchical structure of the hollow micro/nanostructures garnered significant interest due to its ability to offer more active sites, shorten the electron transport path, and increase the contact area with reactants [30,31]. These advantages make the hollow micro/nanostructured $Ni_3Si_2O_5(OH)_4$ a promising candidate for various applications, including photocatalytic reactions and energy-related processes. However, to the best of our knowledge, there are few studies on the synthesis of tubular $Ni_3Si_2O_5(OH)_4$ with a hierarchical structure using natural wollastonite as a template. The use of natural wollastonite minerals offers distinct advantages, such as abundant reserves, easy processing, and low cost, compared to the nanotube templates employed in previous research [32].

Building upon the discussion, our study innovatively explores the utilization of mineral-based CdS for hydrogen evolution, an area with limited reports and potential for improving the hydrogen production rates. Notably, $Ni_3Si_2O_5(OH)_4$, serving as a semiconductor, demonstrates suitable energy levels for the photocatalytic reduction of $CO_2$ to CO [33]. In this context, the preparation of $CdS/Ni_3Si_2O_5(OH)_4$ composite photocatalysts, employing tubular or rod-shaped $Ni_3Si_2O_5(OH)_4$ as the carriers with a hierarchical structure, presents multiple advantages. Specifically, it enhances the separation efficiency of photogenerated carriers through the construction of heterojunctions and improves the dispersion and recyclability of CdS. Consequently, this innovative approach addresses the scarcity of reports on mineral-based CdS for hydrogen evolution and underscores the potential to enhance hydrogen production rates.

Based on the above background, in this paper, the natural wollastonite mineral is used as a template, while the hydrothermal method is employed to in situ synthesize the rod-like $Ni_3Si_2O_5(OH)_4$ composed of nanosheets. Then, the $Ni_3Si_2O_5(OH)_4$ rod is used as a carrier to load CdS particles for constructing a $CdS/Ni_3Si_2O_5(OH)_4$ heterostructure (CdS/NS). In this study, the structure and morphology evolution of $Ni_3Si_2O_5(OH)_4$ rod and CdS/NS are systematically investigated. Meanwhile, the photocatalytic $H_2$ production performance is evaluated, and the performance improvement mechanism is also revealed.

## 2. Results and Discussion

### 2.1. Structure and Morphology of $Ni_3Si_2O_5(OH)_4$

Figure S1 presents the XRD of the $Ni_3Si_2O_5(OH)_4$ samples prepared under different conditions. In the XRD pattern of 1-NS-150-10, weak characteristic peaks were observed at 12.00°, 19.66°, 24.31°, and 60.43°, corresponding to the (002), (110), (004), and (060) crystal planes of $Ni_3Si_2O_5(OH)_4$, respectively. Notably, distinct characteristic peaks of wollastonite were still evident, suggesting an incomplete transformation into $Ni_3Si_2O_5(OH)_4$. When the addition amount of NiCl was increased to Ni:Ca = 3:1 under the same heating conditions (3-NS-150-10), most of the characteristic peaks of wollastonite disappeared, and two new characteristic peaks were seen at 34.05° and 38.36°, representing the (200) and (202) crystal planes of $Ni_3Si_2O_5(OH)_4$, respectively. However, the characteristic diffraction peak at 38.36° deviated from the position of the standard card peak of $Ni_3Si_2O_5(OH)_4$, indicating that, although most of the wollastonite participated in the reaction, the reaction remained incomplete. Furthermore, the crystal structure growth of the $Ni_3Si_2O_5(OH)_4$ nanosheets appeared to be incomplete (Figure S1a). For the samples prepared under a fixed Ni:Ca ratio of 3:1 and at different reaction temperatures and times, the XRD characterization results are presented in Figure S1b. When Ni:Ca = 3:1, and the temperature was raised from 150 °C to 200 °C (3-NS-200-10), a characteristic peak of quartz was observed at 26.5°, indicating that a considerable amount of wollastonite participated in the reaction, resulting in a small portion of quartz impurities from the raw material being exposed on the surface. Additionally, for 3-NS-200-10, the characteristic peak intensity corresponding to the wollastonite (-2-22) facet at 30° was further reduced, and the characteristic peak position of the (202) facet of $Ni_3Si_2O_5(OH)_4$ shifted significantly to the left, approaching that of the corresponding facet in the standard card of $Ni_3Si_2O_5(OH)_4$. This observation suggests that increasing the reaction temperature not only accelerates the reaction rate but also improves

the crystallinity of $Ni_3Si_2O_5(OH)_4$. When the reaction time was further prolonged from 10 h to 24 h (3-NS-200-24), the peak corresponding to wollastonite disappeared completely, and the location of the peak of the (202) crystal plane of $Ni_3Si_2O_5(OH)_4$ continued to shift to the left, reaching 36.65°, which coincided with the location of the peak of the crystal plane in the standard card of $Ni_3Si_2O_5(OH)_4$. This observation indicates that prolonging the reaction time allows for the complete reaction between $Ni^{2+}$ and wollastonite, further enhancing the crystallinity of $Ni_3Si_2O_5(OH)_4$. The above results demonstrate the successful preparation of $Ni_3Si_2O_5(OH)_4$ using wollastonite as a template, and there is no wollastonite phase observed in the sample after the reaction. Furthermore, the reaction temperature and reaction time have a significant effect on the crystallinity of $Ni_3Si_2O_5(OH)_4$, which is crucial for the formation of surface nanosheets and the subsequent loading of CdS.

SEM was utilized to investigate the morphology changes in the $Ni_3Si_2O_5(OH)_4$ prepared under different conditions, and the results are presented in Figure 1. Moreover, Figure S2 displays the SEM images of wollastonite before reaction. In Figure S2, the wollastonite particles are columnar, with smooth surfaces, lengths of about 10–200 μm, and cross-sectional radii of 1–5 μm. Figure 1a shows the SEM image of 1-NS-150-10, where the surface of wollastonite appears rough with few $Ni_3Si_2O_5(OH)_4$ sheets formed on its surface. When the addition amount of NiCl is increased to Ni:Ca = 3:1 under the same heating conditions, more visible nanosheets appear on the surface of wollastonite (Figure 1b). Based on the previous XRD results, it is confirmed that these are $Ni_3Si_2O_5(OH)_4$ nanosheets, suggesting that an increased amount of $Ni^{2+}$ promotes the growth of the $Ni_3Si_2O_5(OH)_4$ nanosheets. When Ni:Ca = 3:1, and the temperature is raised from 150 °C to 200 °C (3-NS-200-10), a dense layer composed of $Ni_3Si_2O_5(OH)_4$ nanosheets is formed on the surface of wollastonite, and the $Ni_3Si_2O_5(OH)_4$ sheets are significantly smaller (Figure 1c), which may be attributed to the fact that the increase in temperature accelerates the nucleation rate of the $Ni_3Si_2O_5(OH)_4$ crystals, thus resulting in a large number of fine $Ni_3Si_2O_5(OH)_4$ nanosheets being generated. So, the final heating temperature chosen was 200 °C. To explore the impact of prolonged heating time on the growth of the layered $Ni_3Si_2O_5(OH)_4$ material, the reaction time was extended from 10 h to 24 h under the heating condition of 200 °C. When the reaction time was further prolonged to 24 h (3-NS-200-24), the small sheets on the surface grew, and eventually, $Ni_3Si_2O_5(OH)_4$ rods (NS) with a nanosheet composition were produced. By comparing the element distribution of the samples before and after the reaction (Figure S3), it was observed that the $Ca^{2+}$ in wollastonite was completely replaced by $Ni^{2+}$. The above results further indicate that the $Ni_3Si_2O_5(OH)_4$ rod with a hierarchical structure is successfully synthesized using natural wollastonite as a template, which provides a prerequisite for loading highly dispersed CdS particles.

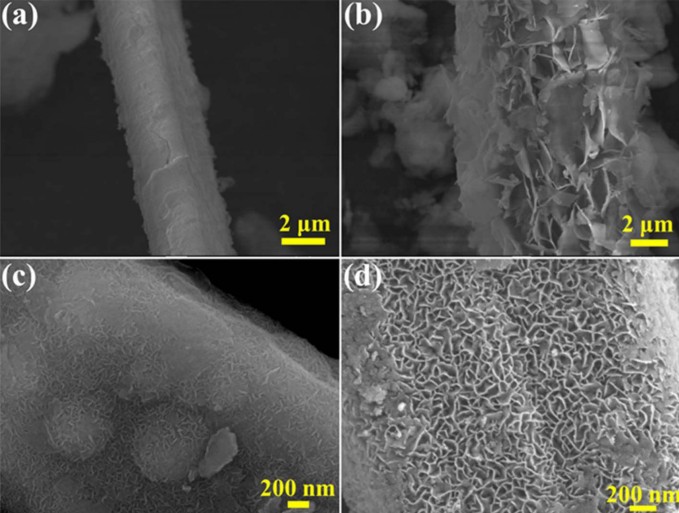

**Figure 1.** SEM of 1-NS-150-10 (**a**), 3-NS-150-10 (**b**), 3-NS-200-10 (**c**), and 3-NS-200-24 (**d**).

## 2.2. Characterizations of CdS/NS

The SEM and TEM images of $Ni_3Si_2O_5(OH)_4$ prepared under optimized conditions (NS), CdS, and CdS/NS are seen in Figure 2. In Figure 2b, there are a large number of nanosheets on the surface of $Ni_3Si_2O_5(OH)_4$ and the surface of smooth nanosheets, while the CdS nanoparticles prepared by chemical deposition suffer from severe inter-particle agglomeration (Figure 2c). As for CdS/NS (Figure 2d,e), compared to $Ni_3Si_2O_5(OH)_4$, the surface of the nanosheets becomes rough where the small particles are uniformly distributed. It can be inferred that these small particles are CdS nanoparticles. Figure 2d1–d5 show the element distribution of CdS/NS corresponding to Figure 2d. It shows that, besides the presence of Ni, O, and Si elements, the distribution range of Cd and S is also aligned with the positions corresponding to CdS/NS, demonstrating a highly uniform distribution. This not only confirms the characteristic elements of $Ni_3Si_2O_5(OH)_4$ (Ni, O, and Si), but also provides visible evidence for the uniform loading of CdS elements on the surface of the $Ni_3Si_2O_5(OH)_4$ nanosheets. To further illustrate the significant improvement in the dispersion of CdS loaded onto the surface of the $Ni_3Si_2O_5(OH)_4$ nanosheets, the morphology of the CdS/NS-mix was studied (Figure 2f). It is found that the CdS/NS-mix exhibits a severe agglomeration of CdS nanoparticles after the simple mixing of $Ni_3Si_2O_5(OH)_4$ and CdS. The aforementioned outcomes demonstrate that the in situ synthesized nanosheets on the $Ni_3Si_2O_5(OH)_4$ surface can considerably enhance the dispersity of CdS, thereby boosting the reaction active sites on CdS.

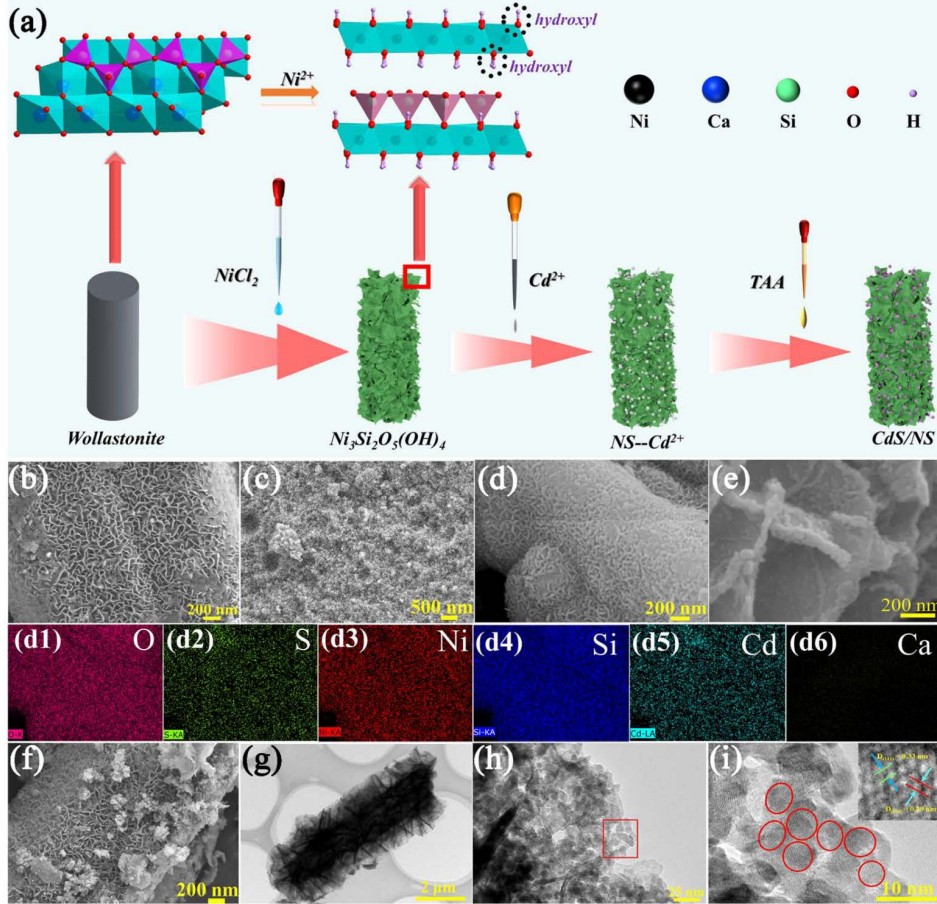

**Figure 2.** The process of synthesizing $Ni_3Si_2O_5(OH)_4$ using wollastonite as the template and preparing CdS/NS (**a**); SEM of $Ni_3Si_2O_5(OH)_4$ (**b**); SEM of CdS (**c**); SEM of CdS/NS (**d,e**): O (d1), Al (d2), Si (d3), Fe (d4), Ti (d5) and Ca (d6) element distributions of CdS/NS (**d**); SEM of CdS/NS-mix (**f**); the TEM of $Ni_3Si_2O_5(OH)_4$ synthesized with wollastonite as the template (**g**); and the TEM of CdS/NS (**h,i**).

Figure 2g–i exhibit the TEM images of $Ni_3Si_2O_5(OH)_4$ and CdS/NS. Figure 2g illustrates that $Ni_3Si_2O_5(OH)_4$ adopts a hollow rod structure composed of nanosheets, indicating the successful in situ synthesis of $Ni_3Si_2O_5(OH)_4$ nanosheets using wollastonite as a template. Figure 2h,i present the HRTEM images of CdS/NS, revealing the uniformly distributed CdS nanoparticles on the surface of the $Ni_3Si_2O_5(OH)_4$ nanosheets. Upon enlarging the red box in Figure 2h to obtain Figure 2i, it is observed that the red area represents CdS nanoparticles with a particle scale of about 2–7 nm. The lattice fringes in the CdS region are magnified, and the lattice spacing of CdS was measured to be 0.33 nm and 0.29 nm, corresponding to the (111) and (200) crystal planes of cubic CdS. The TEM results further prove the successful construction of the CdS/NS composite.

Figure 3a displays the XRD of $Ni_3Si_2O_5(OH)_4$ and CdS/NS. The presence of the characteristic peak of quartz in the patterns of all three samples indicated the existence of an impurity in the wollastonite raw material. In the XRD patterns of CdS/NS, the characteristic peaks of 2θ appearing at 12.00°, 19.66°, 24.31°, 34.05°, 38.36°, and 60.43°, which correspond to the (002), (110), (004), (200), (202), and (060) crystal faces of $Ni_3Si_2O_5(OH)_4$, illustrated that the loading of CdS did not influence the physical phase of $Ni_3Si_2O_5(OH)_4$. The XRD patterns of CdS/NS revealed characteristic peaks at 44.06° and 52.19°, representing the crystal planes of (220) and (311) of CdS, respectively, indicating the successful compounding of CdS and $Ni_3Si_2O_5(OH)_4$. FT-IR analysis was employed to characterize the surface groups of CdS, $Ni_3Si_2O_5(OH)_4$, and CdS/NS, and the results are displayed in Figure 3b. $Ni_3Si_2O_5(OH)_4$ exhibited an absorption peak at 471.41 $cm^{-1}$, accounting for the bending vibration of Si-O. The absorption peak observed at 661.01 $cm^{-1}$ accounted for the Ni-O antisymmetric stretching vibration [34]. Additionally, the absorption peak at 1018 $cm^{-1}$ was ascribed to the Si-O-Si antisymmetric stretching vibration. The absorption peak at 1629.55 $cm^{-1}$ corresponded to the O-H bending vibration of interlayer water [35], while the peak at 3647.06 $cm^{-1}$ was associated with the stretching vibration of O-H between nickel oxide layers [36]. Furthermore, the presence of an absorption band in the range of 3600–3200 $cm^{-1}$ indicated the existence of surface O-H groups [37]. These hydroxyl peaks are likely to facilitate the interaction between $Ni_3Si_2O_5(OH)_4$ and CdS. After loading CdS, a noticeable shift in the absorption band of O-H on the composite surface was observed, indicating a close interaction between CdS and $Ni_3Si_2O_5(OH)_4$ through hydroxyl interactions. This shift further supports the successful combination of CdS and $Ni_3Si_2O_5(OH)_4$ in the composite material [38].

The specific surface area and pore size distribution of wollastonite, $Ni_3Si_2O_5(OH)_4$, and CdS/NS were obtained by a BET analysis. Figure 3c shows the $N_2$ adsorption–desorption isotherms of wollastonite, $Ni_3Si_2O_5(OH)_4$, and CdS/NS. It was found that both $Ni_3Si_2O_5(OH)_4$ and CdS/NS exhibited a type II curve, with a noticeable increase in the adsorption curve at low pressures, indicating the presence of microporous structures in both materials. Figure 3d displays the pore size distribution of wollastonite, $Ni_3Si_2O_5(OH)_4$, and CdS/NS. $Ni_3Si_2O_5(OH)_4$ and CdS/NS both exhibited pores below 2 nm and mesopores above 3 nm. Specifically, the mesopores of $Ni_3Si_2O_5(OH)_4$ were mainly in the size range from 3 nm to 5 nm, while the mesopores of CdS/NS were mainly distributed within the range from 3 nm to 15 nm. The number of micropores below 2 nm and mesopores in the range from 3 nm to 7 nm in $Ni_3Si_2O_5(OH)_4$ was higher compared to CdS/NS, indicating a decrease in the number of these pores after loading CdS onto the surface of the $Ni_3Si_2O_5(OH)_4$ nanosheets. Conversely, the mesoporous distribution of CdS/NS increased significantly in the range from 7 nm to 15 nm, which suggests the growth and development of CdS on the surface of $Ni_3Si_2O_5(OH)_4$, forming a new mesoporous structure. Both $Ni_3Si_2O_5(OH)_4$ and CdS/NS exhibited multi-level pore structures, indicating their strong adsorption capacity. The multi-level pore structure in CdS/NS is beneficial for mass transfer, allowing for the efficient adsorption of reactants onto its surface and the rapid consumption of photogenerated electrons. This leads to an improved reaction efficiency and inhibiting the recombination of photogenerated carriers, ultimately enhancing the photocatalytic activity.

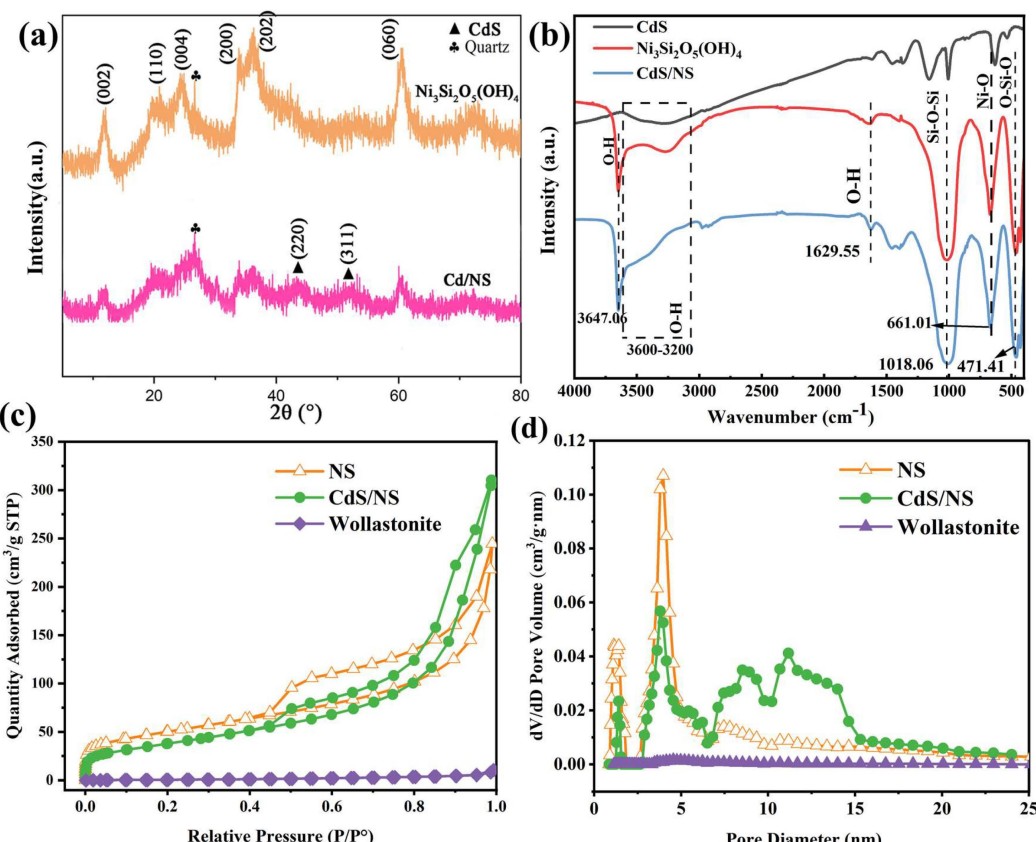

**Figure 3.** The XRD of $Ni_3Si_2O_5(OH)_4$ and CdS/NS (**a**); the FT-IR of $Ni_3Si_2O_5(OH)_4$, CdS, and CdS/NS (**b**); the $N_2$ adsorption and desorption isotherms of $Ni_3Si_2O_5(OH)_4$, CdS, and wollastonite (**c**); and the pore size distribution of $Ni_3Si_2O_5(OH)_4$, CdS, and wollastonite (**d**).

Table S1 reveals that the specific surface area of wollastonite is relatively small, measuring only 2 $m^2$/g, and the pore volume is almost 0 cc/g. In contrast, the $Ni_3Si_2O_5(OH)_4$ with a hierarchical structure, synthesized using wollastonite as a template, exhibits a significantly larger specific surface area of 178.4 $m^2$/g, along with a higher pore volume of 0.276 cc/g. These results indicate that the synthesized $Ni_3Si_2O_5(OH)_4$ demonstrates a considerable increase in specific surface area and developed new pores, further supporting its enhanced performance as a photocatalyst. The specific surface area of CdS/NS was reduced to be 137 $m^2$/g, with a pore volume of 0.4 cc/g. Both $Ni_3Si_2O_5(OH)_4$ and CdS/NS exhibited large specific surface areas, indicating their strong surface activity and adsorption capacity. The decrease in specific surface area and the increase in pore volume of CdS/NS in comparison to $Ni_3Si_2O_5(OH)_4$ suggest that CdS nanoparticles are loaded on the surface of $Ni_3Si_2O_5(OH)_4$ nanosheets, aligning with the findings from the pore size distribution analysis. The results obtained demonstrate that the $Ni_3Si_2O_5(OH)_4$ rod with a hierarchical structure, synthesized using natural wollastonite minerals as a template, exhibited a significant increase in specific surface area and pore structure, which greatly facilitated the enhanced dispersion of CdS. The mechanism by which $Ni_3Si_2O_5(OH)_4$ improved the dispersion of CdS can be described as follows: Initially, $Ni_3Si_2O_5(OH)_4$ adsorbed $Cd^{2+}$ ions onto its lamellar structure due to its large specific surface area. Subsequently, the in situ synthesis of nano-CdS particles occurred on the surface of $Ni_3Si_2O_5(OH)_4$ nanosheets with the addition of $S^{2-}$ ions. The resulting CdS nanoparticles were effectively prevented from agglomerating together, attributed to the adsorption capacity of $Ni_3Si_2O_5(OH)_4$, thus leading to the enhanced dispersion of CdS nanoparticles. These findings provide valuable insights into the unique role of $Ni_3Si_2O_5(OH)_4$ as an effective carrier for improving the dispersion and performance of CdS in photocatalytic applications.

### 2.3. Binding Property of CdS/NS

An XPS analysis was conducted to investigate the valence states of atoms in the samples, including CdS, $Ni_3Si_2O_5(OH)_4$, and CdS/NS, as well as their inner layer electronic binding energy. Figure 4a displays the total XPS spectra, where CdS exhibits characteristic peaks corresponding to Cd 3d and S 2p, and $Ni_3Si_2O_5(OH)_4$ exhibits characteristic peaks of Ni 2p, O 1s, and Si 2p. In addition, CdS/NS exhibits characteristic peaks of Ni 2p, O 1s, Si 2p, Cd 3d, and S 2p, providing evidence that CdS is successfully loaded onto $Ni_3Si_2O_5(OH)_4$.

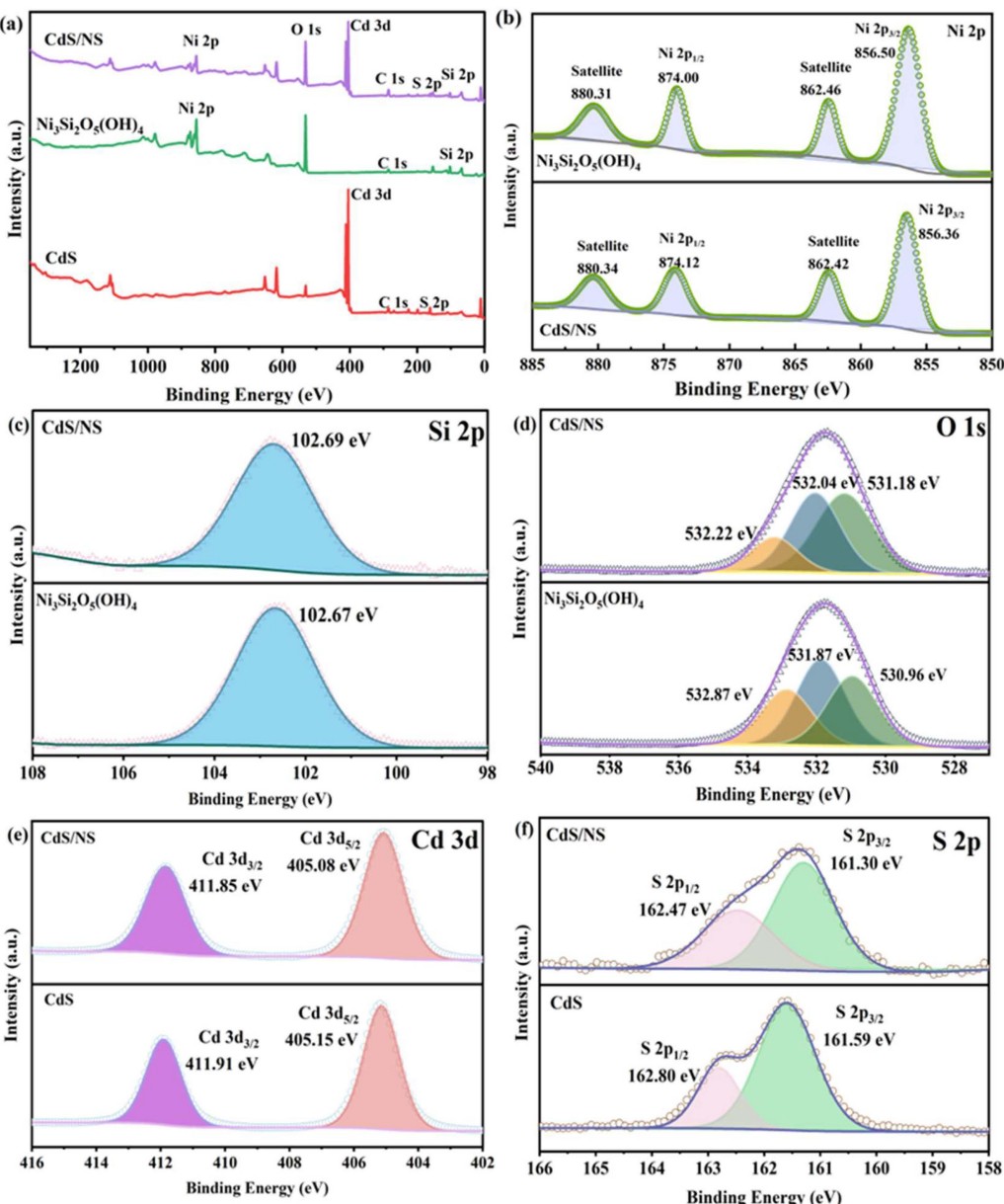

**Figure 4.** The XPS full spectra of CdS, $Ni_3Si_2O_5(OH)_4$, and CdS/NS (**a**); the Ni 2p spectra of $Ni_3Si_2O_5(OH)_4$ and CdS/NS (**b**); the Si 2p spectra of $Ni_3Si_2O_5(OH)_4$ and CdS/NS (**c**); the O 1s spectra of $Ni_3Si_2O_5(OH)_4$ and CdS/NS (**d**); the Cd 3d spectra of CdS and CdS/NS (**e**); and the S 2p spectra of CdS and CdS/NS (**f**).

For $Ni_3Si_2O_5(OH)_4$, the peaks observed in the Ni 2p spectrum at 856.36 eV and 874.00 eV (Figure 4b) corresponded to the characteristic peaks of Ni $2p_{3/2}$ and Ni $2p_{1/2}$, respectively, indicating that the Ni element existed in the form of +2 valence [39,40]. In

the Si 2p spectrum of $Ni_3Si_2O_5(OH)_4$ (Figure 4c), the peak at 102.67 eV was assigned to Si in the silicon oxygen tetrahedron in the metal silicate hydride [41]. Additionally, in the O 1s spectrum of $Ni_3Si_2O_5(OH)_4$ (Figure 4d), the peaks observed at 530.96 eV, 531.87 eV, and 532.87 eV were associated with Ni-O, Si-O, and O-H bonds [42–44], respectively. For CdS, the peaks at 405.15 eV and 411.91 eV in the Cd 3d spectrum (Figure 4e) were the characteristic peaks of Cd $3d_{5/2}$ and Cd $3d_{3/2}$, respectively [45]. Additionally, the peaks observed at 161.59 eV and 162.80 eV in the S 2p spectrum (Figure 4f) belonged to S $2p_{3/2}$ and S $2p_{1/2}$ in the CdS spectrum, respectively [46].

For CdS/NS, the characteristic peaks of Ni $2p_{3/2}$ and Ni $2p_{1/2}$ were detected at 856.50 eV and 874.12 eV and their satellite peaks were found to be 862.42 eV and 880.34 eV, respectively. Notably, the binding energy of Ni $2p_{3/2}$ and Ni $2p_{1/2}$ increased by 0.14 eV and 0.12 eV, respectively (Figure 4b), indicating a reduction in the outer layer charge concentration of Ni after compounding with CdS. After CdS loading, the binding energy of Si 2p remained relatively unchanged at 102.69 eV, indicating that CdS loading on the nanosheets had no significant effect on the Si element (Figure 4c). In the O 1s spectrum (Figure 4d), the O 1s binding energy of Ni-O and Si-O increased by 0.22 eV and 0.21 eV, respectively, while the O 1s binding energy of O-H decreased by 0.66 eV. In the Cd 3d spectrum, the binding energies of the characteristic peaks of Cd $3d_{5/2}$ and Cd $3d_{3/2}$ were 405.08 eV and 411.85 eV, respectively, showing a reduction of 0.07 eV and 0.06 eV compared to pure CdS (Figure 4e). The binding energy of the characteristic peaks of S $2p_{3/2}$ and S $2p_{1/2}$ were 161.30 eV and 162.47 eV, respectively, which decreased by 0.29 eV and 0.33 eV (Figure 4f). In summary, it is hypothesized that CdS and $Ni_3Si_2O_5(OH)_4$ are bonded by a dehydroxylation reaction to form Ni-O-S bonds. The formation of these chemical bonds contributed to the good stability of the composite photocatalyst. Additionally, the binding energy shifts of different elements provide further insights. The binding energy of S 2p in CdS decreased significantly, suggesting an increase in the outer charge density. On the other hand, the binding energy of O 1s and Ni 2p in Ni-O of $Ni_3Si_2O_5(OH)_4$ increased, implying a decrease in the charge density. These results suggest the formation of a heterojunction between $Ni_3Si_2O_5(OH)_4$ and CdS.

### 2.4. Photocatalytic H$_2$ Production Performance of CdS/NS

The pH affects the release of sulfur ions from thiourea as a sulfur source, which in turn affects the production of CdS; so, the effect of the pH on the performance was explored. Figure 5a illustrates the $H_2$ generation performance of CdS/NS-Z (where Z represents the pH value) prepared under different pH conditions. The $H_2$ production efficiency of CdS/NS-10 was measured at 2.38 mmol $h^{-1}$ $g^{-1}$. As the pH increased to 12, the $H_2$ production efficiency of CdS/NS-12 decreased to 2.00 mmol $h^{-1}$ $g^{-1}$. Further adjusting the pH to 14 led to a significant decrease in the $H_2$ production efficiency of CdS/NS-14 to a minimum of 0.42 mmol $h^{-1}$ $g^{-1}$. These comparisons demonstrate that the efficiency of $H_2$ production diminishes as pH increases. This trend can be attributed to the accelerated release of a substantial amount of $S^{2-}$ into the solution during thiourea decomposition at higher pH levels, consequently promoting the rapid formation of CdS with $Cd^{2+}$. Furthermore, the particle size tends to increase with the rise in the pH, leading to an augmented density and a reduced availability of active sites. Consequently, this leads to a decrease in the efficiency of hydrogen production [47,48]. In Figure 5b, the influence of different $Cd^{2+}$ concentrations on the $H_2$ generation performance of the prepared samples is depicted. The $H_2$ generation performance of CdS/NS exhibited an initial increase followed by a subsequent decrease with the rise in the $Cd^{2+}$ concentration. At a $Cd^{2+}$ concentration of 400 ppm, the $H_2$ generation efficiency of CdS/NS reached its peak value of 4.05 mmol $h^{-1}$ $g^{-1}$. In Figure 5c, the impact of the Ni dosage on the preparation of $Ni_3Si_2O_5(OH)_4$ using the template method is illustrated. The performance of CdS/NS demonstrated an enhancement with the increase in the Ni content. Particularly, when the Ni: Ca ratio was 3:1, the $H_2$ generation efficiency of the prepared sample reached 4.00 mmol $h^{-1}$ $g^{-1}$. This improvement can be accounted for the higher concentration of

$Ni^{2+}$, which boosts the growth of $Ni_3Si_2O_5(OH)_4$ nanosheets, leading to the formation of $Ni_3Si_2O_5(OH)_4$ nanosheets with improved crystallinity.

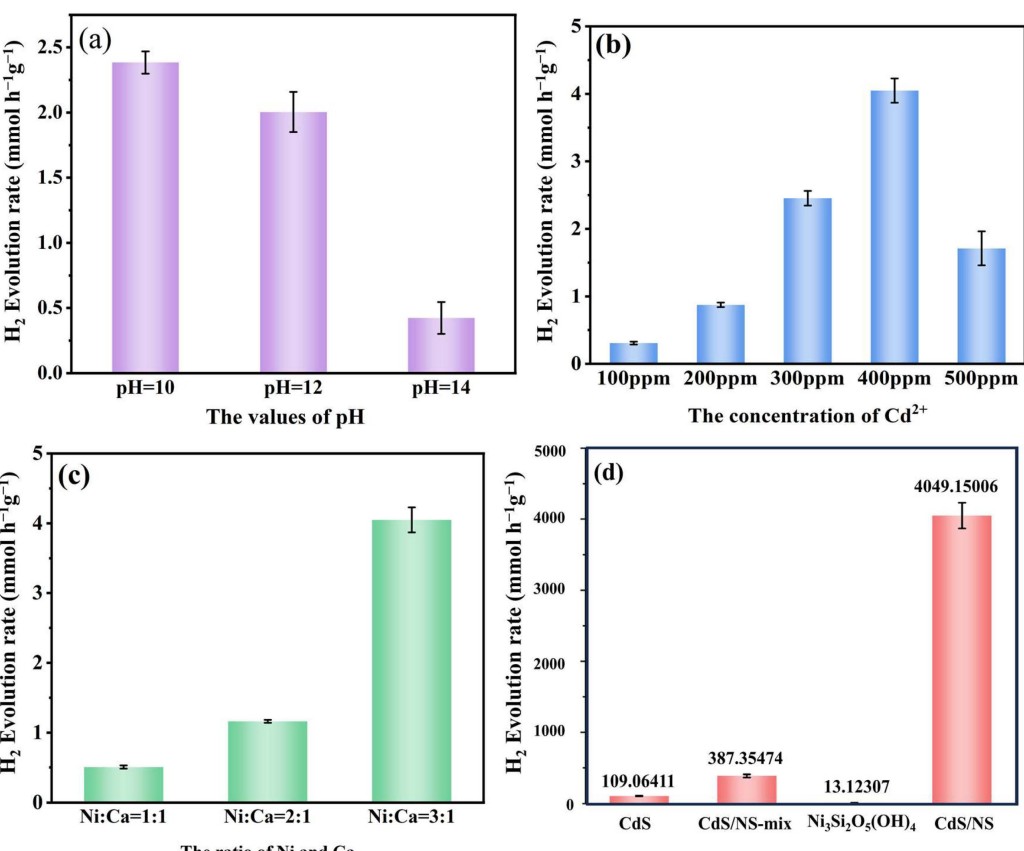

**Figure 5.** Photocatalytic $H_2$ production performance of CdS/NS under different preparation conditions: different pH values (**a**), the concentration of $Cd^{2+}$ (**b**), the mole ratio of $Ni^{2+}$ and $Ca^{2+}$ in wollastonite (**c**), and the photocatalytic $H_2$ production performance of CdS, CdS/NS-mix, $Ni_3Si_2O_5(OH)_4$, and CdS/NS (**d**).

In Figure 5d, the photocatalytic $H_2$ generation performance of CdS, $Ni_3Si_2O_5(OH)_4$, CdS/NS, and CdS/NS-mix is presented. $Ni_3Si_2O_5(OH)_4$ displayed a relatively low $H_2$ generation efficiency of only 0.01 mmol $h^{-1}$ $g^{-1}$, indicating its limited photocatalytic activity. CdS showed a slightly higher $H_2$ generation efficiency of 0.11 mmol $h^{-1}$ $g^{-1}$, while CdS/NS-mix demonstrated a further improvement with an efficiency of 0.39 mmol $h^{-1}$ $g^{-1}$. However, the most significant enhancement was observed in CdS/NS, which exhibited an impressive $H_2$ generation efficiency of 4.5 mmol $h^{-1}$ $g^{-1}$. This value was 40 times higher than that of CdS alone and 10.4 times higher than that of CdS/NS-mix, respectively. These results conclusively demonstrate that the loading of CdS on $Ni_3Si_2O_5(OH)_4$ nanosheets can dramatically improve the photocatalytic $H_2$ generation performance. Based on the testing and characterization, it can be inferred that the improved dispersion of CdS and the formation of heterojunctions are the key factors contributing to the exceptional photocatalytic hydrogen evolution efficiency of CdS/NS. Table 1 compares the efficiency of NS/CdS with other photocatalysts used for photocatalytic hydrogen production.

**Table 1.** Comparison of the photocatalytic hydrogen production rate of CdS/NS with other catalysts.

| Catalyst | Light Source | $M_{catalyst}$ | $V_{reaction\ solution}$ | Hydrogen Production Rate | Ref. |
|---|---|---|---|---|---|
| Pd–CdS/g-C$_3$N$_4$ | 300 W Xe ($\lambda$ > 420 nm) | 50 mg | 100 mL | 293.0 µmol·g$^{-1}$ h$^{-1}$ | [49] |
| CdS-Co$_3$O$_4$ | 350 W Xe ($\lambda$ > 420 nm) | 0.05 g | 80 mL | 150.7 µmol h$^{-1}$ | [50] |
| CdSe/CuInS$_2$ | 300 W Xe | 10 mg | 100 mL | 10610.37 µmol·g$^{-1}$ h$^{-1}$ | [51] |
| CdSe/CdS | 300W Xe ($\lambda$ > 420 nm) | 10 mg | 100 mL | 16.03 mmol·g$^{-1}$ h$^{-1}$ | [52] |
| CdS/Ni-MOF | 300W Xe ($\lambda$ > 420 nm) | 30 mg | 60 mL | 7.83 mmol·g$^{-1}$ h$^{-1}$ | [53] |
| Mn$_{0.2}$Cd$_{0.8}$S/CoFe$_2$O$_4$/rGO | 300W Xe ($\lambda$ > 420 nm) | 50 mg | 100 mL | 133.5 µmol·g$^{-1}$ h$^{-1}$ | [54] |
| Ag@CoFe$_2$O$_4$/g-C$_3$N$_4$ | 300W Xe ($\lambda$ > 420 nm) | 5 mg | 50 mL | 335 µmol·g$^{-1}$ h$^{-1}$ | [55] |
| g-C$_3$N$_4$@ZnIn$_2$S$_4$ | 5 W blue LED ($\lambda$max = 420 nm) | 50 mg | 50 mL | 2377.6 µmol·g$^{-1}$ h$^{-1}$ | [56] |
| R-TiO$_2$/n-TiO$_2$ | 300W Xe ($\lambda$ > 420 nm) | 30 mg | 100 mL | 4.05 mmol h$^{-1}$ g$^{-1}$ | This study |

In Figure 6, the performance of the CdS/NS photocatalytic H$_2$ production cycle is illustrated. During the cycling experiments, the initial hour showed a relatively low H$_2$ production, possibly due to CdS/NS having a certain pore distribution that led to H$_2$ storage in the pores. However, the subsequent hourly H$_2$ production remained remarkably stable. Specifically, the hydrogen production during the first 5 h light was 15.65 mmol/g, which was similar to the second 5 h light with 15.56 mmol/g. The third 5 h light cycle yielded 15.31 mmol/g, and the fourth 5 h light cycle produced 15.24 mmol/g. These consistent H$_2$ production values over the four cycles indicate that there is no significant decrease in the photocatalytic H$_2$ generation performance of CdS/NS. These findings demonstrate that the heterojunction photocatalyst composed of CdS and Ni$_3$Si$_2$O$_5$(OH)$_4$ exhibits an excellent recycling performance and demonstrates a good stability throughout the cyclic H$_2$ production process.

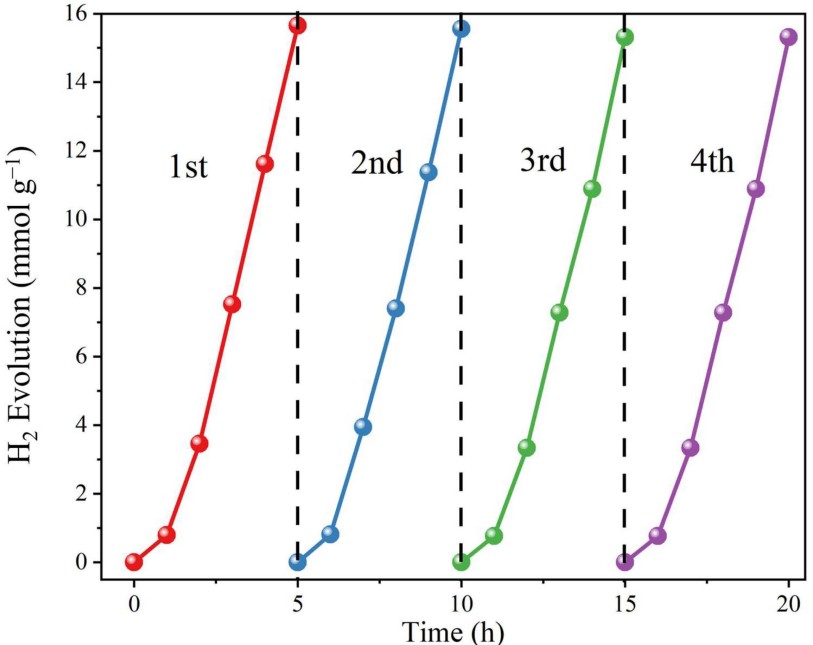

**Figure 6.** The cyclic photocatalytic hydrogen evolution of CdS/NS.

### 2.5. Photocatalytic Mechanism

In Figure 7a, the results obtained using photoluminescence spectroscopy (PL) to assess the recombination rate of photoinduced electrons and holes are presented. The PL peak intensity of CdS/NS-mix was observed to be the highest, suggesting a low efficiency of photogenerated carrier separation in CdS/NS-mix. On the other hand, the PL peak intensity of CdS/NS was the lowest, indicating that the in situ synthesis of CdS on the surface of the $Ni_3Si_2O_5(OH)_4$ nanosheets effectively reduced the recombination rate of the photoinduced carriers, which was attributed to the successful construction of a heterojunction between CdS and $Ni_3Si_2O_5(OH)_4$. The photocurrent intensity of the samples under simulated sunlight was determined to further validate the separation effect of the photoinduced carriers during the photocatalytic process, and the results are presented in Figure 7b. Compared to CdS/NS-mix, the photocurrent of CdS/NS exhibited a significant increase, indicating a decrease in the recombination efficiency of the photogenerated electrons and holes. The findings are in agreement with the results of the PL tests and further support the successful construction of heterojunctions between CdS and $Ni_3Si_2O_5(OH)_4$, which effectively promotes the efficient separation and utilization of photogenerated carriers in the CdS/NS composite photocatalyst. Figure 7c displays the impedance measurements of CdS, $Ni_3Si_2O_5(OH)_4$, CdS/NS-mix, and CdS/NS. The highest impedance observed for CdS was comparable to that of CdS/NS-mix, indicating that the simple mixing of CdS and NS did not enhance the photogenerated carrier separation efficiency of CdS. This result further verifies the successful construction of a heterojunction between CdS and NS in the composite photocatalyst. On the other hand, CdS/NS exhibited the smallest impedance, indicating that the composite photocatalyst had a higher photogenerated carrier separation efficiency. These findings suggest that the successful heterostructure formation between CdS and $Ni_3Si_2O_5(OH)_4$ contributed to the efficient separation of photoinduced carriers in CdS/NS [57]. In summary, the combination of photoluminescence spectroscopy, photocurrent intensity measurement, and impedance analysis provided solid evidence for the efficient photogenerated carrier separation in CdS/NS, further confirming the significant enhancement in the photocatalytic performance due to the successful construction of a heterojunction between CdS and $Ni_3Si_2O_5(OH)_4$ in the composite photocatalyst.

Figure 8a shows the UV–visible diffuse reflectance spectra of CdS, $Ni_3Si_2O_5(OH)_4$, CdS/NS-mix, and CdS/NS. The bandgap width was calculated using the Kubelka–Munk (K-M) function (Figure 8b) [58]. In Figure 8a, it can be observed that the photoresponse range of $Ni_3Si_2O_5(OH)_4$ was relatively narrow, indicating its limited light absorbance in the visible light range. In contrast, CdS exhibited a broader photoresponse range, implying its higher light absorption efficiency in the visible light range. When comparing the spectra of CdS/NS-mix and CdS/NS, both showed a blueshift compared to CdS, indicating a modification in their electronic band structures [59,60] However, the blueshift in the spectrum of CdS/NS was more pronounced than that of CdS/NS-mix. The significant blueshift in CdS/NS was ascribed to the construction of a Z-Scheme heterojunction between CdS and $Ni_3Si_2O_5(OH)_4$ in the composite photocatalyst. The Z-Scheme heterojunction in CdS/NS allows for high-efficient charge separation and transfer, reducing the recombination rate of photoinduced electron–hole pairs [61]. Therefore, CdS/NS exhibited enhanced photocatalytic activity compared to CdS and CdS/NS-mix. The modification in the spectra and the formation of a Z-Scheme heterojunction further confirmed the successful construction of heterojunctions between CdS and $Ni_3Si_2O_5(OH)_4$ in CdS/NS, which contributed to the observed improvement in the photocatalytic performance. In Figure 8b, the bandgap width of $Ni_3Si_2O_5(OH)_4$ was measured to be 2.72 eV, indicating its semiconducting nature with a relatively wide bandgap. Both CdS and CdS/NS-mix showed a bandgap width of 2.27 eV, consistent with their typical semiconducting properties. Notably, the bandgap width of CdS/NS was increased to 2.44 eV, demonstrating the successful construction of a heterojunction between CdS and NS in the composite material. This increase in bandgap width is attributed to the interactions at the interface of CdS and NS, which cause a variation in the electronic band structure and bandgap of the composite photocatalyst. The

formation of heterojunctions in CdS/NS is responsible for the observed improvements in the photocatalytic performance, as it promotes efficient charge separation and transfer, reducing the recombination of photogenerated electron–hole pairs [62]. The increased bandgap width in CdS/NS further confirms the effective construction of a heterojunction between CdS and $Ni_3Si_2O_5(OH)_4$, enhancing the photocatalytic activity of CdS/NS for $H_2$ evolution.

In Figure 8c,d, the flat band potentials of CdS and $Ni_3Si_2O_5(OH)_4$ were measured as $-0.95$ V and $-0.82$ V, respectively. These flat band potentials can be converted into the conduction band potentials relative to the standard hydrogen electrode through calculations. Combining the front bandgap widths, the valence band potentials of the samples can also be determined. For $Ni_3Si_2O_5(OH)_4$, the conduction band potential was found to be $-0.62$ eV, and the valence band potential was 2.1 eV. In the case of CdS, the conduction band potential was $-0.75$ eV, and the valence band potential was 1.52 eV. The results demonstrate that CdS and $Ni_3Si_2O_5(OH)_4$ are provided with the conditions for the construction of heterojunctions. The XPS analysis indicates that Ni and O outer electrons in Ni-O-S tend to lose electrons after loading, while S tends to gain electrons, which suggests that, during the photoreaction process, the photogenerated electrons at the conduction band position of $Ni_3Si_2O_5(OH)_4$ transfer to the valence band of S in CdS through Ni-O-S and subsequently combine with photogenerated holes. These findings support the hypothesis that the photocatalytic activity enhancement observed in CdS/NS is owed to the high-efficient charge carrier separation and transfer facilitated via the heterojunction between CdS and $Ni_3Si_2O_5(OH)_4$.

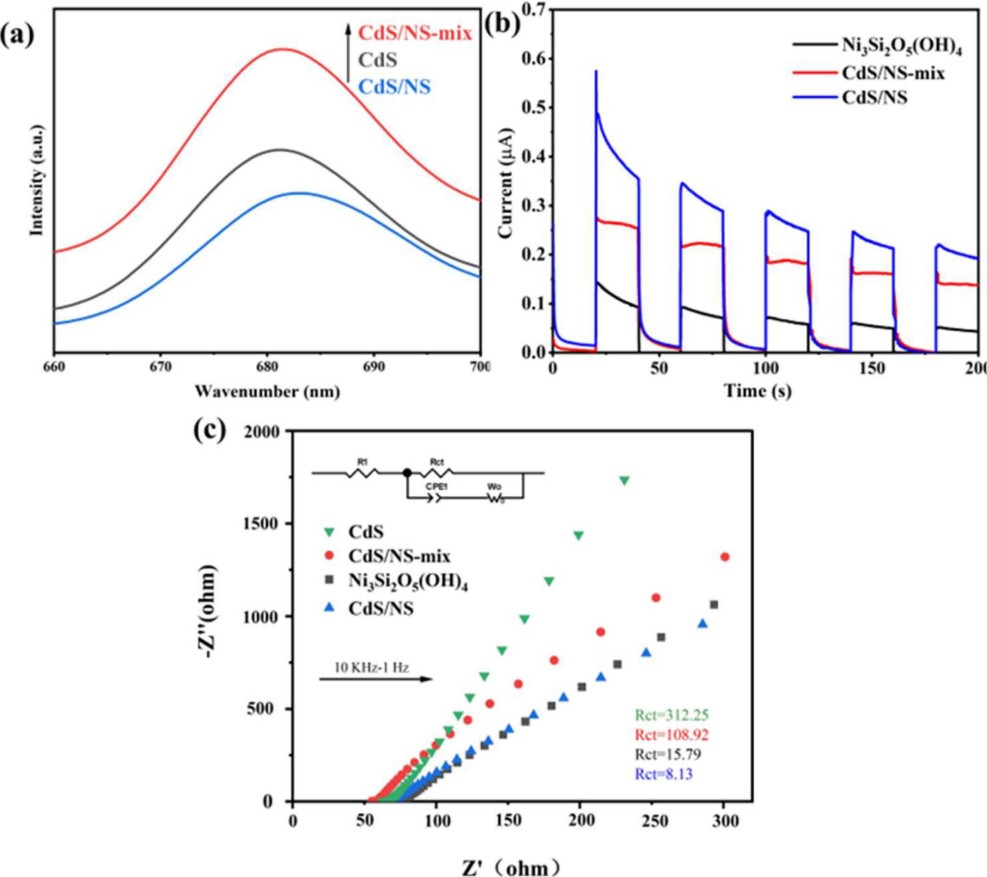

**Figure 7.** PL spectra (**a**), the transient photocurrent (**b**), and the EIS spectra (**c**).

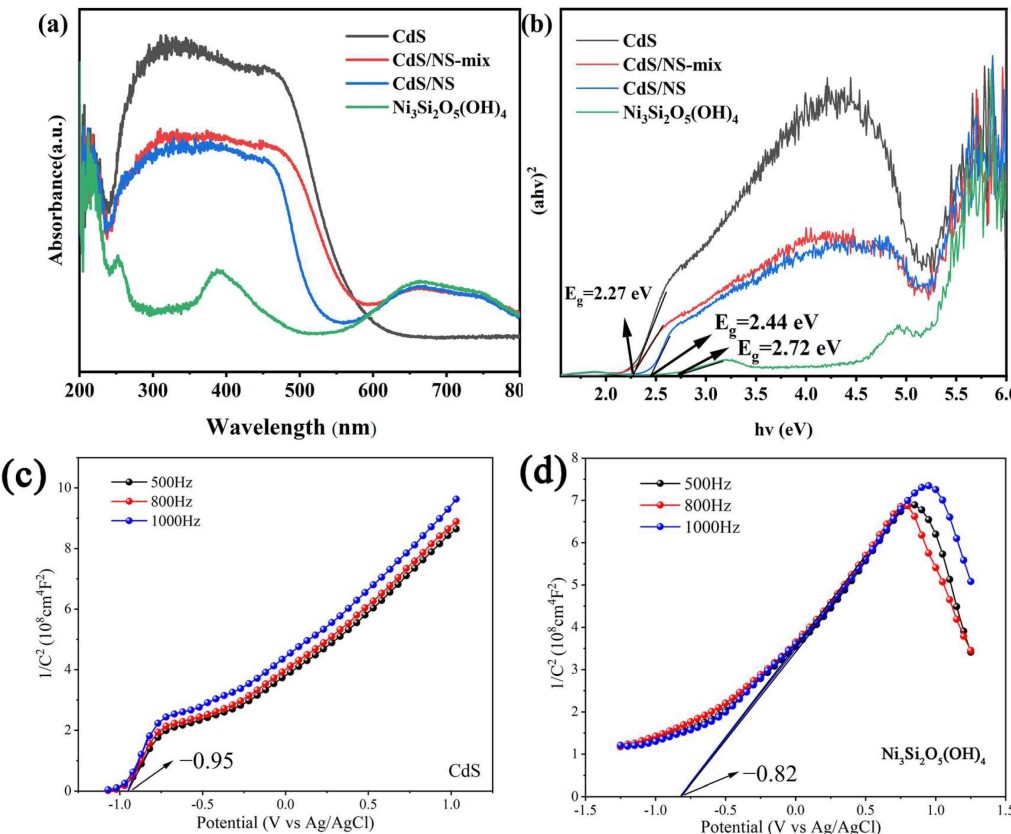

**Figure 8.** UV–visible diffuse reflectance spectra of CdS, $Ni_3Si_2O_5(OH)_4$, CdS/NS-mix, and CdS/NS (**a**); the band gap of CdS, $Ni_3Si_2O_5(OH)_4$, CdS/NS-mix, and CdS/NS (**b**); and the Mott–Schottky of CdS and $Ni_3Si_2O_5(OH)_4$ (**c**,**d**).

Based on the comprehensive test and calculation results, the mechanism of how CdS/NS improves the photocatalytic $H_2$ generation performance can be ascribed to the following points: Firstly, the hierarchical structure of $Ni_3Si_2O_5(OH)_4$ effectively disperses the CdS nanoparticles on its surface, resulting in more exposed active sites of CdS. This enhanced dispersion increases the available surface area for photocatalytic reactions, leading to an improved photocatalytic efficiency. Moreover, CdS/NS possesses a large specific surface area and multi-level pore structure, which enhances its adsorption capacity for reactants onto the surface. This adsorption effect accelerates the reaction rate, thereby enhancing the overall photocatalytic performance. More importantly, the construction of a heterostructure between CdS nanoparticles and $Ni_3Si_2O_5(OH)_4$ is a key factor in boosting the photogenerated carrier separation of the composite photocatalyst, thereby promoting the utilization of these charge carriers for photocatalytic hydrogen production. To summarize, the combination of a hierarchical structure, large specific surface area, multi-level pore structure, and heterostructure formation results in a high-efficient and stable CdS/NS photocatalyst for $H_2$ generation. The mechanisms are visually illustrated in Figure 9.

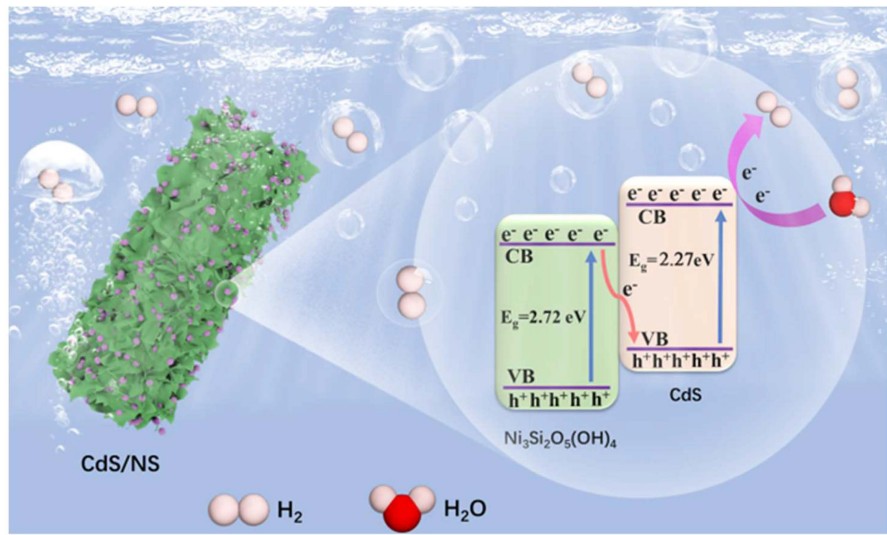

**Figure 9.** Mechanism of the CdS/NS photocatalytic decomposition of water for hydrogen production.

## 3. Materials and Methods

### 3.1. Raw Materials and Reagents

The wollastonite raw material was obtained as a powder from Dalian Global Minerals Co., Ltd., Dalian, China. The source of cadmium was cadmium nitrate, supplied by Shandong Xiya Chemical Co., Ltd. (Yantai, China) Nickel (II) chloride hexahydrate, obtained from Shanghai Aladdin Technology Co., Ltd. (Shanghai, China), served as the nickel source. Thiourea (TAA), provided by Shanghai Macklin Biochemical Technology Co., Ltd. (Shanghai, China), was used as the sulfur source. Sodium hydroxide (NaOH), from Beijing Lanyi Chemical Products Co., Ltd. (Beijing, China), was utilized to adjust the pH of the solution. Anhydrous ethanol, which was provided by Shanghai Macklin Biochemical Technology Co., Ltd. (Shanghai, China), and deionized water served as the solvents for the experiment.

### 3.2. Preparation of $Ni_3Si_2O_5(OH)_4$ Using Wollastonite as the Template

The preparation process of $Ni_3Si_2O_5(OH)_4$ is shown in Figure 2a. First, 25.8 mmoL of $NiCl_2$ was dissolved in 70 mL of deionized water to prepare the $NiCl_2$ solution. Subsequently, 1 g of wollastonite was added to the $NiCl_2$ solution to achieve different mole ratios of $Ni^{2+}$ and $Ca^{2+}$ in wollastonite (1:1 and 3:1, respectively). The mixture was stirred for 15 min to obtain a suspension. The obtained suspension was then transferred to a reactor and subjected to hydrothermal reactions at different temperatures (150 °C and 200 °C) for varying durations (10 h and 24 h). After the reaction, the container was chilled to room temperature, and the sample was centrifuged, washed with deionized water five times, and subsequently dried at 80 °C for 12 h to yield the $Ni_3Si_2O_5(OH)_4$ rods composed of nanosheets with wollastonite as the template. For the convenience of comparison, the $Ni_3Si_2O_5(OH)_4$ rods, obtained with different mole ratios of Ni and Ca in wollastonite (1:1 and 3:1, respectively), different temperatures (150 °C and 200 °C), and different reaction times (10 h and 24 h), denoted as X-NS-M-Y, where X represents the mole ratios of $Ni^{2+}$ and $Ca^{2+}$ in wollastonite, M represents the reaction temperatures, and Y represents the reaction time for the hydrothermal reaction.

### 3.3. Preparation of $CdS/Ni_3Si_2O_5(OH)_4$

The preparation process of $CdS/Ni_3Si_2O_5(OH)_4$ is shown in Figure 2a. To prepare the Cd $(NO_3)_2$ solution, Cd $(NO_3)_2$ was added to a solvent comprising anhydrous ethanol and deionized water, resulting in a specific concentration of the Cd $(NO_3)_2$ solution. Subsequently, 100 mg of $Ni_3Si_2O_5(OH)_4$ was placed in 150 mL of the Cd $(NO_3)_2$ solution, and the hybrid mixture was stirred for 1 h to obtain a suspension. We dissolved 60 mg

of thiourea in 5 mL of deionized water and set it aside for later use. Thiourea was then added to the suspension and stirred for an additional 15 min. Concurrently, 1 mol/L of NaOH solution was gradually added dropwise to adjust the pH value. We controlled the pH within the range from 9 to 14. The reaction vessel was moved to an oil bath at 80 °C for 5 h. After the reaction, the suspension was centrifuged and separated more than three times with anhydrous ethanol and deionized water until the supernatant was neutral. Finally, the obtained $Ni_3Si_2O_5(OH)_4$ rod-loaded CdS composite photocatalyst (CdS/NS) was dried in an oven at 75 °C for 6 h. A pure CdS was also prepared using the same conditions. For studying the interface interaction of CdS/NS, a mixture of CdS and NS (CdS/NS-mix) was prepared. The information about the characterization details and photocatalytic $H_2$ evolution performance of the as-prepared samples can be seen in the Supplementary Materials.

*3.4. Photocatalytic $H_2$ Evolution Performance*

Firstly, 30 mg of the sample was placed inside a quartz reaction vessel. Subsequently, 90 mL of deionized water and 10 mL of lactic acid (used as a sacrificial agent) were added to the reaction vessel. The container was then purged with argon gas for 40 min to eliminate any air present. Once the air was completely removed, the reactor was transferred to a photoreactor for the light reaction, with a 300 W xenon lamp serving as the light source. Hourly syringe samples were taken and injected into a gas chromatography system (GC9790 Plus, Fuli Instruments, Wenling, Zhejiang, China) to measure the content of $H_2$ produced.

**4. Conclusions**

The study successfully synthesized hollow micro/nanostructured nickel silicate hydroxide ($Ni_3Si_2O_5(OH)_4$) by using natural wollastonite mineral as a template. The resulting $Ni_3Si_2O_5(OH)_4$ had a rod-like structure with nanosheets on its surface, exhibiting a significantly larger specific surface area (80 times) compared to raw wollastonite ($2\ m^2/g$). CdS nanoparticles were uniformly loaded onto the nanosheets of $Ni_3Si_2O_5(OH)_4$, showing an excellent dispersibility. The CdS/NS photocatalyst in the presence of lactic acid as the sacrificial agent demonstrated an impressive $H_2$ production rate of 4.05 mmol $h^{-1}$ $g^{-1}$. This efficiency was around 40 times and 405 times higher than those of pure CdS and $Ni_3Si_2O_5(OH)_4$, respectively. CdS/NS maintained a consistent $H_2$ production efficiency even after four cycles, indicating an excellent stability. The successful construction of a heterojunction between $Ni_3Si_2O_5(OH)_4$ and CdS contributed to an improved photogenerated carrier separation efficiency in the composite photocatalyst, leading to an enhanced photocatalytic $H_2$ production performance. This work not only provides a novel choice to synthesize materials with micro/nanostructures using a natural mineral as the template, but also presents a new strategy to address challenges in heterojunction photocatalysts for hydrogen production.

**Supplementary Materials:** The following supporting information can be downloaded at: https://www.mdpi.com/article/10.3390/catal14030183/s1. Section S1: Test and characterizations; Section 2: Photocatalytic $H_2$ evolution performance; Figure S1: The XRD patterns of $Ni_3Si_2O_5(OH)_4$ prepared under different conditions: (a) different mole ratios of Ni and Ca in wollastonite and (b) different reaction temperatures and times; Figure S2: The SEM of wollastonite before the reaction; Figure S3: The SEM of wollastonite (a): Ca (a1) and Ni (a2) element distribution of wollastonite (a); the SEM of 3-NS-200-24 (b): Ca (b1) and Ni (b2) element distribution of 3-NS-200-24 (b); Figure S4: The band structures of CdS and $Ni_3Si_2O_5(OH)_4$; Table S1: Specific surface area and pore volume of wollastonite, $Ni_3Si_2O_5(OH)_4$, and CdS/NS.

**Author Contributions:** Conceptualization, J.M., R.Z. and D.C.; methodology, J.M.; validation, Y.T. and R.Z.; investigation, J.M. and R.Z.; resources, R.M.; writing—original draft preparation, J.M.; writing—review and editing, Y.T. and R.Z.; visualization, J.M. and D.C.; supervision, H.D.; project administration, H.D.; funding acquisition, H.D. All authors have read and agreed to the published version of the manuscript.

**Funding:** This research was funded by the Fundamental Research Funds for the Central Universities of China, grant number 292018301.

**Data Availability Statement:** The data are available within the article.

**Conflicts of Interest:** The authors declare no conflicts of interest.

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
