# Peer review of "Natural Wollastonite-Derived Two-Dimensional Nanosheet Ni3Si2O5(OH)4 as a Novel Carrier of CdS for Efficient Photocatalytic H2 Generation"

_catalysts, doi:10.3390/catal14030183_

Round 1
Reviewer 1 Report
Comments and Suggestions for Authors
I accept the manuscript to be published in journal of catalysts after minor revision. The research work is interesting. However, some comments must taken in consideration before publication
1-Fig. 7a that construct PL analysis of the as-synthesized samples must be measured and illustrated in more better form as the sample scattering is high and the authors write (a.u.) but there is value in y-axis
2-What is the experimental evidence that proof the construction of Z-scheme mechanism
3-More recent approach was based on S-scheme mechanism which is more precise in predicating the charge transportation between two semiconductors are the author test this new mechanism instead of Z-scheme mechanism
4-EDX analysis can be added to the revised manuscript
5-A comparative study table for comparing the photocatalytic efficiency of the as-synthesized sample with the famous photocatalyst in the recent literature can add to the revised manuscript
Comments on the Quality of English LanguageThe English language is appropriate
Reviewer 2 Report
Comments and Suggestions for Authors
The work represents a completed study on photocatalytic hydrogen evolution using a CdS-based photocatalyst. However, there are some remarks:
- authors need to emphasize the purpose and novelty of the work in the introduction
- The signatures on the graphs are very small
- The methodology for conducting a photocatalytic experiment should be in the main text of the article, not in supporting
- the abstract must indicate which system (electron donor) was used for hydrogen evolution
- Has the calibration of the binding energy scale in XPS spectra been performed?
- What is the maximum of apparent quantum yield achieved in this work?
- what are the units of absorbance in fig. 8a?
Reviewer 3 Report
Comments and Suggestions for Authors
The present work concerned with formation of CdS/Ni3Si2O5(OH)4 composites using double stage process that included hydrothermal process followed by water bath assisred process. The obtained performce of the hybrid catalysts toward the H2 production is good. However, there are several major comment that need to be considered before publications. Comments and recommendations are mentioned below:
1) in the introduction section, H2 performance of CdS-Minerals based composite should be reported to reveal demonstrate the novelty of the present work.
2) Materials and method section shoukd be transferred after the introduction section as famaliar. Authors should illustrate why they change two parameters in the preparation at the same time temperature and reaction time.
3) The used condentration of TAA and pH range used i the prepartik should be mentioned in details.
4) The H2 generation performance at pH below 10 is required to demonstrate the optimum performance at which pH.
5) How many times samles were measured for H2 at different pH.
6) The dependence of H2-generation on pH still need more explanation and should be supported with reference/ or TEM image at different pH should be provided to reveal the agglomaration at high pH.
7) The discussion section is lake of supporting references. So, more demonstration is necessary with suitable references.
8) The PL emission should be measured from 500 to 700 nm.
9) The Raman spectra should be provided becuase FTIR spectra could not verify the active vibration related to CdS because of the bands overlaps with Ni3Si2O5(OH)4.
10) The fitting parameters of EIS spectra should be reported to demonstrate the change in the interfacial resistance (Rct). How, it seems that Ni3SiO7(OH)4 has lower EIS arc thand the best sample CdS/NS composite.
11) there is calculation mistkes in the optical band gap. Authors should correctly use kubelka-munk function and mentione the used equation in this section.
12) the fitting of EIS results based on Mott-schoottky including donor concentration (ND), and flat band potentiol was not reported. So, full analysis is required.
Comments on the Quality of English LanguageLanguage need moderate modification.
Round 2
Reviewer 3 Report
Comments and Suggestions for Authors
Authors have improved their manuscript based on the suggested recommendations. Hence, I think it is suitable for publication in the present form.
Comments on the Quality of English LanguageEnglish language of manuscript is ok.
Author Response
We thank the reviewer for all the useful suggestions on this manuscript and we have revised the English language expressions in the manuscript.